# High-Resolution FTIR Spectroscopy of CH_3_F: Global Effective Hamiltonian Analysis of the Ground State and the 2*ν*_3_, *ν*_3_ + *ν*_6_, and 2*ν*_6_ Bands

**DOI:** 10.3390/molecules30224389

**Published:** 2025-11-13

**Authors:** Hazem Ziadi, Michaël Rey, Alexandre Voute, Jeanne Tison, Bruno Grouiez, Laurent Manceron, Vincent Boudon, Hassen Aroui, Maud Rotger

**Affiliations:** 1Groupe de Spectrométrie Moléculaire et Atmosphérique (GSMA), UMR 7331 CNRS, Université de Reims Champagne-Ardenne, Moulin de la Housse B.P. 1039, F-51687 Reims Cedex 2, France; 2Laboratoire de Spectroscopie et de Dynamique Moléculaire, Université de Tunis, École Nationale Supérieure d’Ingénieurs de Tunis, 5 Av. Taha Hussein, Tunis 1008, Tunisia; 3AILES Beamline, Synchrotron SOLEIL, L’Orme des Merisiers, Départementale 128, F-91190 Saint-Aubin, France; 4Laboratoire Interuniversitaire des Systèmes Atmosphériques (LISA), Université Paris Cité, Université Paris-Est Créteil, CNRS, F-75013 Paris, France; 5Laboratoire Interdisciplinaire Carnot de Bourgogne Europe (ICB), UMR 6303 CNRS—Université de Bourgogne, 9 Av. A. Savary, BP 47870, F-21078 Dijon Cedex, France

**Keywords:** methyl fluoride, hydrofluorocarbons (HFCs), high-resolution spectroscopy, high-order contact transformations, vibration–rotation states

## Abstract

High-resolution Fourier transform infrared (FTIR) spectra of methyl fluoride (CH_3_F) were recorded in the mid- and far-infrared regions using the Bruker IFS 125HR spectrometers at GSMA (Reims, France) and at the SOLEIL synchrotron facility (Saint-Aubin, France). The measurements cover both the pure rotational transitions of the ground state (10–100 cm^−1^) and the vibrational triad region (1950–2450 cm^−1^), which includes the 2$\nu_{3}$, $\nu_{3}+\nu_{6}$, and 2$\nu_{6}$ bands. Spectra were recorded under various pressure conditions to optimize line visibility, with a high resolution. Line assignments were performed using predictions from the tensorial effective Hamiltonian implemented in the MIRS package, together with a newly developed automated assignment tool, SpectraMatcher, which facilitates line matching and discrimination of CH_3_F transitions from overlapping CO_2_ features. More than 5000 transitions (up to J=52 in the ground state and up to J=45 in the triad and K=19) were assigned and included in a global fit. The sixth-order tensorial effective Hamiltonian model yielded excellent agreement with experiment, with root mean square (RMS) deviations better than 7 × 10^−4^ cm^−1^ across all regions. This paper presents the first continuous rovibrational study of CH_3_F over both the triad and far-infrared ground state regions. The improved accuracy from previous studies stems from the improved set of effective Hamiltonian parameters which will also form a good basis from future applications in atmospheric modelling and spectroscopic databases.

## 1. Introduction

High-resolution infrared spectroscopy of polyatomic molecules is important because it provides key details concerning the molecular structure and dynamics of the molecules themselves and offers important spectroscopic data to be used in modeling atmospheric and astrophysical applications. Accurate line lists of the vibrational and rotational transitions are vital for modeling planetary and astrophysical spectra, and for monitoring the atmosphere on Earth.

Methyl fluoride (CH_3_F) has been the subject of numerous high-resolution spectroscopic studies in recent decades, motivated by its fundamental interest as a prototypical symmetric top molecule and its relevance to atmospheric and planetary science [1,2,3]. Extensive experimental and theoretical investigations have been conducted to generate accurate spectroscopic databases for this molecule, covering both the far-infrared [4,5] and mid-infrared regions [6,7,8,9,10,11,12,13,14,15,16,17]. In recent years, particular attention has been devoted to the strong $\nu_{3}$ fundamental band [18], the $\nu_{3}+\nu_{6}$ combination band [19] and the $\nu_{2}$/$\nu_{5}$ dyad [20]. These studies contribute to refining effective Hamiltonian models and dipole moment parameters, thereby improving the predictive capabilities of databases such as HITRAN [21] and GEISA [22]. Methyl fluoride (CH_3_F) is a symmetric-top molecule of $C_{3v}$ symmetry with six vibrational modes, among which the $\nu_{3}$ (C-F stretching) and $\nu_{6}$ (deformation) modes play a key role in the mid-infrared region. Owing to its relatively simple structure, CH_3_F has long served as a benchmark system for testing effective Hamiltonian models and dipole moment expansions in symmetric top molecules [23,24].

Several previous studies have focused on individual vibrational bands of CH_3_F. Early high-resolution investigations of the 2$\nu_{3}$ overtone band by Faust et al. [9] provided assignments up to J=35, though difficulties were reported in resolving low-*K* doublets. Nakagawa et al. [25] analyzed the $\nu_{3}+\nu_{6}$ band using Watson’s Hamiltonian formalism, deriving molecular constants but without a global treatment of the polyad structure. In the far-infrared region, Papousek et al. [4] measured pure rotational transitions up to J=58, establishing accurate ground-state parameters. However, these works were limited to individual bands or restricted quantum numbers, and a consistent global treatment of the triad system (2$\nu_{3}$, $\nu_{3}+\nu_{6}$, 2$\nu_{6}$), together with the ground state has remained lacking.

The present work provides a comprehensive rovibrational analysis of CH_3_F, focusing on the triad region (2$\nu_{3}$, $\nu_{3}+\nu_{6}$, 2$\nu_{6}$) within the 1950–2450 cm^−1^ spectral domain, as well as the pure rotational spectrum in the far-infrared (20–100 cm^−1^). High-resolution FTIR spectra were recorded at the GSMA laboratory (Reims, France) and at the SOLEIL synchrotron facility (Saint-Aubin, France), under various pressure conditions optimized for each spectral region. Assignments were performed using a combination of effective Hamiltonian predictions implemented in the MIRS package [23] and automated matching routines developed in our SpectraMatcher software. A global fit of more than 5000 transitions was carried out using a sixth-order tensorial effective Hamiltonian, yielding a $10^{-3}$ cm^−1^ accuracy across all regions studied.

This work builds on previous research by proposing, for the first time, a coherent description of the ground state and triad regions (2$\nu_{3}$, $\nu_{3}$ + $\nu_{6}$, 2$\nu_{6}$). We thus demonstrate the validity and robustness of the effective tensor Hamiltonian approach for CH_3_F, confirmed by ab initio calculations, and provide a new set of spectroscopic parameters for future modeling and databases.

## 2. Effective Hamiltonian Formalism

Methyl fluoride (CH_3_F) is a symmetric-top molecule belonging to the $C_{3v}$ point group. Its vibrational energy levels can be grouped into polyads (Figure 1), which collect states strongly coupled through anharmonic and Coriolis interactions. This work aims at providing a comprehensive rovibrational analysis of methyl fluoride (CH_3_F), including both the ground state and the interacting vibrational states ($v_{3}=2$, $v_{3}=v_{6}=1$, $v_{6}=2$). The experimental dataset combines high-resolution spectra in the far-infrared (10−100 cm^−1^), probing pure rotational transitions in the ground vibrational state, and in the mid-infrared spectra in the 1950–2450 cm^−1^ region, probing the triad. The band centers are located at 2097.93 cm^−1^ for 2$\nu_{3}$, 2221.80 cm^−1^ for $\nu_{3}+\nu_{6}$, and 2324.26 cm^−1^ for 2$\nu_{6}$. Because these three bands are strongly interacting, they must be treated simultaneously within a global rovibrational model. The ground-state rotational spectrum (10–100 cm^−1^) was also analyzed to refine the ground-state parameters.

To extend the rovibrational analysis of experimental spectra, our CH_3_F model was based on a full rovibrational Hamiltonian [26]. The numerical approach proposed recently in Ref. [27] was employed to generate an initial set of ab initio effective Hamiltonian parameters, using a refined version of the potential energy surface (PES) of Ref. [26]. To this end, the nuclear-motion Hamiltonian was expressed in irreducible tensor operators for a full account of symmetry properties [28]. The PES was also validated using a variational calculation compared to experimental data, in particular to pure rotational transitions up to J=5.

The effective rovibrational Hamiltonian adapted to the polyad structure of the CH_3_F molecule, constructed as the sum of contributions appropriate to the various polyads, is expressed as follows:(1)$$\begin{array}{ccc} H & = & \sum_{k}H_{k}=H_{0}+H_{1}+H_{2}+\ldots \\ & = & H_{\left(GS\right)}+H_{\left(\nu_{3}/\nu_{6}\right)}+H_{\left(\nu_{2}/\nu_{5}\right)}+H_{\left(2\nu_{3}/\nu_{3}+\nu_{6}/2\nu_{6}\right)}\\ & & +H_{\left(Polyad4\right)}+H_{\left(Polyad5\right)}+H_{\left(Polyad6\right)} \end{array}$$ Each term in Equation (Equation 1) writes as a linear combination of rovibrational irreducible tensor operators, expressed as:(2)$$\begin{array}{ccc} H_{\left(\mathrm{Polyad}\right)} & = & \sum_{\mathrm{all}\;\mathrm{indices}}t_{\left\{n_{s}\right\}\left\{m_{s}\right\}}^{\Omega \left(K,n\Gamma_{r}\right)\Gamma_{1}\Gamma_{2}\Gamma_{v}}T_{\left\{n_{s}\right\}\left\{m_{s}\right\}}^{\Omega \left(K,n\Gamma_{r}\right)\Gamma_{1}\Gamma_{2}\Gamma_{v}}\\ & = & t_{\left\{n_{1}n_{2}\ldots \right\}\left\{m_{1}m_{2}\ldots \right\}}^{\Omega \left(K,n\Gamma_{r}\right)\Gamma_{1}\Gamma_{2}\Gamma_{v}}\beta {\left(R^{\Omega (K,n\Gamma_{r})}{⊗}^{\epsilon }V_{\left\{n_{1}n_{2}\ldots \right\}\left\{m_{1}m_{2}\ldots \right\}}^{\Gamma_{1}\Gamma_{2}\Gamma_{v}}\right)}^{\left(A_{1}\right)} \end{array}$$ In this formulation, *V*, *R* and *T* denote the vibrational, rotational and rovibrational tensor operators, while *t* represents the effective parameters to be adjusted in the fitting procedure. The recursive coupling of the creation and annihilation operators associated with the six normal modes of CH_3_F enables the construction of vibrational operators in a tensorial form. $\Gamma_{1}$ and $\Gamma_{2}$ stand for the vibrational symmetries, ϵ is the parity in the conjugate momenta such that $\epsilon ={\left(-1\right)}^{\Omega }$ where Ω specifies the degree in the angular momentum components $J_{x}$, $J_{y}$ and $J_{z}$. The lower indices $n_{i}$ and $m_{i}$ indicate the powers of the vibrational creation and annihilation operators. *K* is the rank of the tensor in O(3). $\Gamma_{r}$ and $\Gamma_{v}$ represent, respectively, the rotational and vibrational symmetries in the $C_{3v}$ point group ($A_{1}$, $A_{2}$, and *E*). β is a normalization factor [23] ensuring consistency with the zero-order spectroscopic constants. Note that the same tensorial formulation applies to the construction of the dipole moment, with operators of symmetry $A_{2}$ and *E*.

## 3. Assignment and Results

Using a vibrational extrapolation scheme adapted to the $C_{3v}$ point group, implemented in the MIRS package [23], the spectra were calculated by means of the effective Hamiltonian parameters of the polyad $P_{0}$ (ground state) and of the triad ($P_{3}$). The sixth-order effective Hamiltonian model employed in this work enabled to achieve an accuracy comparable to that of the experiments.

As an initial step, we used the parameters derived from ab initio calculations and converted into the MIRS formalism. These parameters yielded very good preliminary simulations across all spectral regions studied, including the ground state and the 2$\nu_{3}$, $\nu_{3}+\nu_{6}$, and 2$\nu_{6}$ bands. This facilitated the first round of assignments in the *P*, *Q*, and *R* branches across the different spectral regions.

Our home-made SpectraMatcher software was employed for the first time to visualize and assign experimental spectra. It enables the simultaneous display of the experimental CH_3_F spectrum together with a synthetic CO_2_ absorption spectrum from the HITRAN 2020 database [21], which is essential for distinguishing CH_3_F lines from overlapping CO_2_ features. The software can also generate synthetic CH_3_F spectra based on MIRS predictions, by convolving the calculated line lists with a Gaussian instrumental profile and applying an intensity cut-off to filter out weak transitions. SpectraMatcher automates the assignment process by computing the distance between observed CH_3_F transitions and the corresponding theoretical MIRS predictions, while performing the same comparison for HITRAN CO_2_ lines. In cases where CH_3_F transitions coincided with CO_2_ features, special care was taken during the assignment process to avoid misidentification.

As shown in Table 1, different spectra were selected for the line position analysis. For the $2\nu_{3}$ band, the spectrum recorded at a pressure of 0.09 mbar was used, while for $\nu_{3}+\nu_{6}$ the 1 mbar spectrum was retained. For $2\nu_{6}$, the spectrum recorded at a pressure of 2.56 mbar was selected. For the far-infrared region, all spectra listed in Table 2 were included in the analysis.

It is worth noting that the $\nu_{3}+\nu_{6}$ spectrum presented additional difficulties due to strong CO_2_ absorption in this spectral region.

In such cases, the doublet was treated as a single transition, with its measured intensity taken as twice that of one component. This situation occurs systematically in the 2$\nu_{3}$, 2$\nu_{6}$ and rotational spectra, where all K=3p doublets remain unresolved at our instrumental resolution. In contrast, for the $\nu_{3}+\nu_{6}$ band, about 86 doublets were sufficiently resolved to separate the $A_{1}$ and $A_{2}$ components, which were then matched individually. These resolved transitions belong mainly to the K=3 manifolds of the ^*P*^*P*, ^*P*^*Q*, ^*P*^*R*, ^*R*^*P*, ^*R*^*Q*, and ^*R*^*R* sub-branches. This partial resolution in the $\nu_{3}+\nu_{6}$ region provided valuable additional constraints for the fit of the effective Hamiltonian parameters.

Figure 2 and Figure 3 highlight chosen sections of the comparison between the experimental spectrum and the simulated spectrum showing the standard quantum assignment from SpectraMatcher. Figure 2 shows ground state rotational and $2\nu_{3}$ transitions whereas Figure 3 focuses specifically on $\nu_{3}+\nu_{6}$ and $2\nu_{6}$ bands and has the calculated line intensities represented by black sticks; the K=0 and K=3p transitions are shown in magenta and blue, respectively.

The fitting process was done piecewise to allow for parameter stability. We started with the $\nu_{3}+\nu_{6}$ band-release and adjusted all pertinent parameters for that band. Next, we examined the 2$\nu_{3}$ band: its own parameters were also released and fitted as a single isolated band, but then, we gradually released and fitted the interaction parameters between 2$\nu_{3}$ and $\nu_{3}+\nu_{6}$ once these two bands were stabilized with the same level of fitting. The 2$\nu_{6}$ band was next in line for analysis. Since it had limited intensity and very strong perturbations (mostly from $\nu_{3}+\nu_{2}$ and $\nu_{3}+\nu_{5}$), we just released its own parameters first while ignoring the interaction parameters with 2$\nu_{3}$ and $\nu_{3}+\nu_{6}$ at this time. Once all three bands (2$\nu_{3}$, $\nu_{3}+\nu_{6}$, 2$\nu_{6}$) parameters were stable, we moved onto the refinement of the ground state parameters via the fitting of the far-IR data. In the final phase, we fit a global data set of all the transitions between the three bands presented in the ground state simultaneously for all parameters in the effective Hamiltonian including the interactions.

At the beginning of the assignment process, an intensity cut-off of 1 × 10^−24^ cm/mol was applied consistently. This cut-off enabled SpectraMatcher to automatically identify CH_3_F lines while eliminating nearly all contributions from noise and was particularly effective in distinguishing the CH_3_F lines from CO_2_ lines in the $\nu_{3}+\nu_{6}$ and 2$\nu_{6}$ regions. Next, both automatic and manual assigned CH_3_F lines were saved for later adjustment of line positions using MIRS via automatic fitting.

For the $\nu_{3}+\nu_{6}$ band, the process culminated in the assignment of 3101 lines, of which 2500 were ultimately included in the fit of 18 effective Hamiltonian parameters, yielding an RMS deviation of 7.34 × 10^−4^ cm^−1^.

For the 2$\nu_{3}$ band and the ground state, the assignments were more straightforward: the predicted line positions matched very well with observation, and the same cut-off strategy ensured rapid and unambiguous line identification. In the earlier high-resolution study of Faust et al. [9], the rotational structure was identified up to J=35 with *K* values up to 12, although most of the K=0,1,2 doublets remained unresolved under their experimental conditions. By contrast, thanks to the maximum resolution of our Bruker IFS 125HR spectra (0.003 cm^−1^), we were able to resolve the K=2 doublets and to extend the assignments up to J=45 and K=15. A total of 1060 lines were included in the fit of the 2$\nu_{3}$ band, with an RMS deviation of 6.74 × 10^−4^ cm^−1^. This improvement provided a denser and more accurate set of transitions for the effective Hamiltonian analysis compared with previous studies.

For the ground-state rotational spectrum, our assignments extended significantly beyond those reported by Papoušek et al. [4], who measured transitions up to J=58 at an unapodized resolution of 0.0019 cm^−1^. In the present work, more than 800 pure rotational transitions were assigned up to J=52. However, as in the earlier study, the K=0 and K=1 doublets could not be resolved, even at our improved spectral resolution, similarly to the situation observed in the 2$\nu_{3}$ band. This confirms that, despite enhanced sensitivity and extended assignments, the resolution of these low-*K* doublets remains intrinsically challenging. In total, 849 rotational transitions were fitted, yielding an RMS deviation of 3.56 × 10^−4^ cm^−1^.

In contrast, the 2$\nu_{6}$ band presented a bigger challenge. This overtone is weak and suffers from strong perturbations with nearby combination bands, in particular $\nu_{3}+\nu_{2}$ and $\nu_{3}+\nu_{5}$. Nevertheless, 831 transitions were successfully assigned, yielding an RMS deviation of 7.52 × 10^−4^ cm^−1^. Despite these difficulties, the combined use of MIRS predictions and SpectraMatcher visualization enabled to obtain a consistent set of low-*K* assignments, sufficient to derive reliable effective Hamiltonian parameters for this band.

A representative sample of the electronic Appendix A, showing the observed and assigned transitions of CH_3_F in the far-infrared region around 79 cm^−1^ and in the $\nu_{3}+\nu_{6}$ band between 2222 and 2223 cm^−1^, is presented in Table 3. Table 4 compares the effective Hamiltonian fit results obtained in this work with those reported by Khan et al. [18]. While the Khan et al. [18] performed a global fit including all CH_3_F polyads, from the ground state to higher vibrational states, the present work focuses specifically on the ground state and the 2$\nu_{3}$, $\nu_{3}+\nu_{6}$, and 2$\nu_{6}$ states, which had not previously been analyzed with such precision.

This approach produces a significant enhancement in fit quality for these specific states. As evidenced by the reduced RMS residuals from Khan’s results [18], fit quality is also shown in the normalized residuals shown in Figure 4 (given as the residuals between the observed line positions and the calculated line positions for the ground state and all three triad bands). The residuals are generally well-centered about the zero value, with no systematic deviation, which indicates the accuracy and consistency of the fit proposed in this study. Due to the large number of transitions considered in the fit, particularly both the combined and overtone bands have low residuals, which corroborates the viability of the present method.

To sum up, the assignment process was relatively easy for the ground state, and for the 2$\nu_{3}$ band, it involved more complexity for the $\nu_{3}+\nu_{6}$ because of CO_2_ lines, and for the 2$\nu_{6}$ band, it was substantially more difficult because of strong perturbations from neighboring vibrational states. The successive fitting process enabled stable convergence for the effective Hamiltonian parameters for the entire set of assigned transitions, with 5304 lines fitted to better than $10^{-3}$ cm^−1^ accuracy. Adjusted parameters of the effective Hamiltonian are given in the Appendix A.

## 4. Experimental Details

High-resolution Fourier transform infrared (FTIR) spectra of methyl fluoride (CH_3_F) were recorded at the GSMA laboratory in Reims (France) and at the SOLEIL synchrotron facility (Saint-Aubin, France). Two complementary sets of measurements were performed in order to probe both the triad (2$\nu_{3}$, $\nu_{3}+\nu_{6}$, 2$\nu_{6}$) region in the mid-infrared and the pure rotational spectrum in the far-infrared.

### 4.1. Spectra Recorded at GSMA, Reims

High-resolution absorption spectra of CH_3_F in the triad region (1950–2450 cm^−1^) were recorded with the Bruker IFS 125 HR Fourier transform spectrometer at GSMA (Reims). The interferometer was equipped with a KBr beamsplitter, a globar infrared source, and a liquid-nitrogen-cooled photo-conductive HgCdTe (MCT) detector. The cell temperature was maintained at 294 K, and the interferograms were processed with the OPUS Bruker software [29]. The pressure inside the absorption cells was monitored with a 10 Torr calibrated Baratron gauge (Leybold CTR101N with an accuracy of 0.2%). Unless otherwise stated, the CH_3_F gas sample was supplied by Linde Electronics with a certified purity of 99%. The interferograms were Fourier transformed without any apodization function (Boxcar option) and using a zero-filling factor of 8. All the measurements carried out at Reims were taken at a spectral resolution of 0.003 cm^−1^.

#### 4.1.1. 2$\nu_{3}$ Band (1950–2100 ${\mathrm{cm}}^{-1}$)

Two spectra were recorded at pressures of 0.09 and 0.40 mbar using a vertical multi-pass cell with an effective path length of 3.2 m. Figure 5 shows an overview of the 2$\nu_{3}$ band.

#### 4.1.2. $\nu_{3}+\nu_{6}$ Band (2100–2350 ${\mathrm{cm}}^{-1}$)

Two absorption spectra were measured at pressures of 1 and 3 mbar using a 2 m horizontal White-type multi-pass cell with an effective path length of 8.264 m. Because of the weak intensity of the $\nu_{3}+\nu_{6}$ band, this long optical path was required. In this case, the CH_3_F sample was purchased from Sigma Aldrich (purity 97.5%). Strong CO_2_ lines, present in this spectral region, were used to calibrate the wave number scale against the HITRAN database [21], yielding an accuracy of about 2.3 × 10^−4^ cm^−1^. Figure 6 shows an overview of the $\nu_{3}+\nu_{6}$ region.

#### 4.1.3. 2$\nu_{6}$ Band (2250–2450 ${\mathrm{cm}}^{-1}$)

Two spectra were recorded at pressures of 1.00 and 2.56 mbar using the vertical multi-pass cell with an extended optical path length of about 38 m. Because of the weak intensity of this overtone and its overlap with other vibrational systems ($\nu_{6}+\nu_{2}$, $\nu_{6}+\nu_{5}$), the assignment required careful analysis. The main experimental conditions for the Reims spectra are summarized in Table 1.

### 4.2. Spectra Recorded at SOLEIL Synchrotron

Complementary measurements were performed at the AILES beam line of the SOLEIL synchrotron (Saint-Aubin, France) to record far-infrared spectra of CH_3_F in the 10–100 cm^−1^ range at a resolution higher than in previous works [4,5]. Leveraging the high intensity and brilliance of the synchrotron source in the infrared improved the signal-to-noise ratio and dynamic range, enabling the observation and reliable assignment of higher-*J* ground-state rotational transitions and resulting in more accurate spectroscopic parameters. We used the Bruker IFS 125 HR spectrometer coupled with a 20 cm white cell providing a 84.9 cm optical path, with a multilayer Mylar beamsplitter (6 μm). For the lowest range (10–40 cm^−1^), we used silicon bolometers cooled with liquid helium at 1.6 K, and a bolometer at 4 K for the highest range (40–100 cm^−1^). The pressure of CH_3_F sample (Linde Electronics, 99%) was between 0.07 and 0.59 mbar. The sample spectra were measured at a resolution of 0.001 cm^−1^. The interferograms were Fourier transformed using different apodization functions adapted to each spectral domain.

For the 40–100 cm^−1^ region (4 K bolometer), the spectra were processed without apodization (Boxcar option), which preserves the instrumental resolution. For the 10–40 cm^−1^ region (1.6 K bolometer), both the Boxcar and the Blackman–Harris 3-terms apodization functions were used in a complementary way. The Boxcar function, while noisier, preserves the maximum resolution and was helpful to distinguish weak transitions from noise. The Blackman–Harris function slightly reduces the spectral resolution but significantly improves the signal-to-noise ratio by suppressing side lobes. This allows for more reliable identification of low-*J* rotational lines. This combined approach proved to be essential in this frequency range, where the CH_3_F transitions are weak and affected by noise. In all cases, a zero-filling factor of 8 was applied for smooth interpolation between sample points of the spectra (see Figure 7). The main experimental conditions are summarized in Table 2.

## 5. Conclusions

In this work, we have presented a comprehensive high-resolution spectroscopic study of methyl fluoride (CH_3_F), combining new FTIR measurements recorded at GSMA and at the SOLEIL synchrotron with a global tensorial effective Hamiltonian analysis. The assigned transitions cover both the pure rotational spectrum in the far-infrared region and the triad region including the 2$\nu_{3}$, $\nu_{3}+\nu_{6}$, and 2$\nu_{6}$ bands. Using the home-made SpectraMatcher tool in combination with MIRS simulations, more than 5000 transitions were assigned up to J=52 for the ground state and up to J=45 for the triad. The final global fit, based on a sixth-order tensorial effective Hamiltonian, reproduces the experimental line positions with an overall RMS = 7.10 × 10^−4^ cm^−1^.

This work offers the most precise and consistent rovibrational description of CH_3_F to date, and marks a significant improvement over previous analyses. The new refined set of effective Hamiltonian parameters provides a solid basis for not only the reliable predictions of line positions in the regions that were considered in this paper, but also for reliable predictions for future extensions of the analysis to higher polyads and hot bands. Nonetheless, the value of these results extends beyond reliable predictions: they will be freely available for spectroscopic databases, and for practical applications for atmosphere and astrophysics.

## Figures and Tables

**Figure 1 molecules-30-04389-f001:**
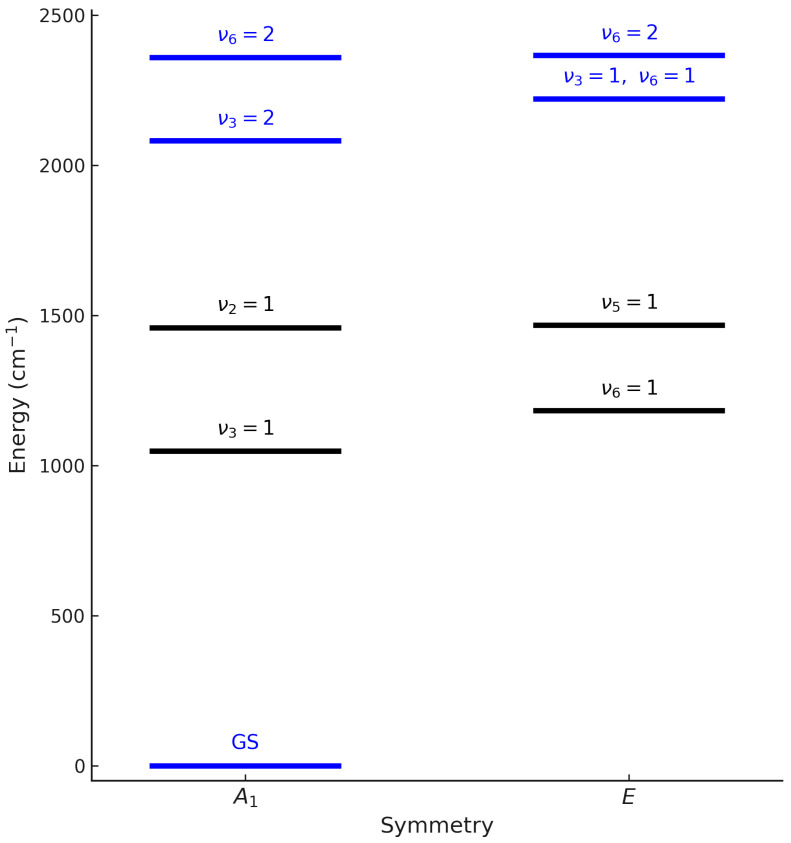
Simplified energy diagram of CH_3_F rovibrational levels by symmetry (A_1_ and E). The levels shown in blue (GS, $v_{3}=2$, $v_{3}=v_{6}=1$, $v_{6}=2$) correspond to the ground state and the target triad, which are the focus of our experimental and theoretical investigation. The other levels are displayed for reference.

**Figure 2 molecules-30-04389-f002:**
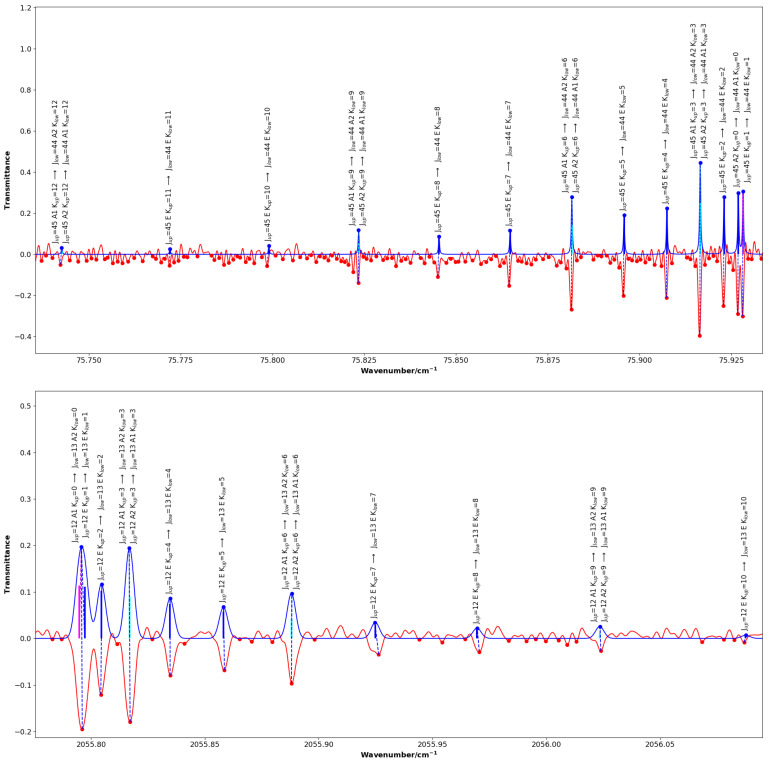
Portions of the assigned far-infrared ground state rotational spectrum (**top**) and mid-infrared $2\nu_{3}$ band spectrum (**bottom**) of CH_3_F, the simulated spectra are blue and point up, while the experimental spectra are red and point down (transmission). For the ground state, transitions with J=44 and *K* ranging from 0 to 12 are displayed: the K=0 and K=1 components are fully resolved, whereas the K=3p lines remain unresolved. For the $2\nu_{3}$ band, transitions with J=13 and *K* up to 10 are shown: the K=0 and K=1 components are blended. Calculated line intensities are represented by vertical black sticks within the profiles, with K=0 transitions shown in magenta and K=3p transitions in blue. The labels displayed above each assigned line indicate the quantum numbers *J* and *K*, as well as the symmetries of the upper and lower states.

**Figure 3 molecules-30-04389-f003:**
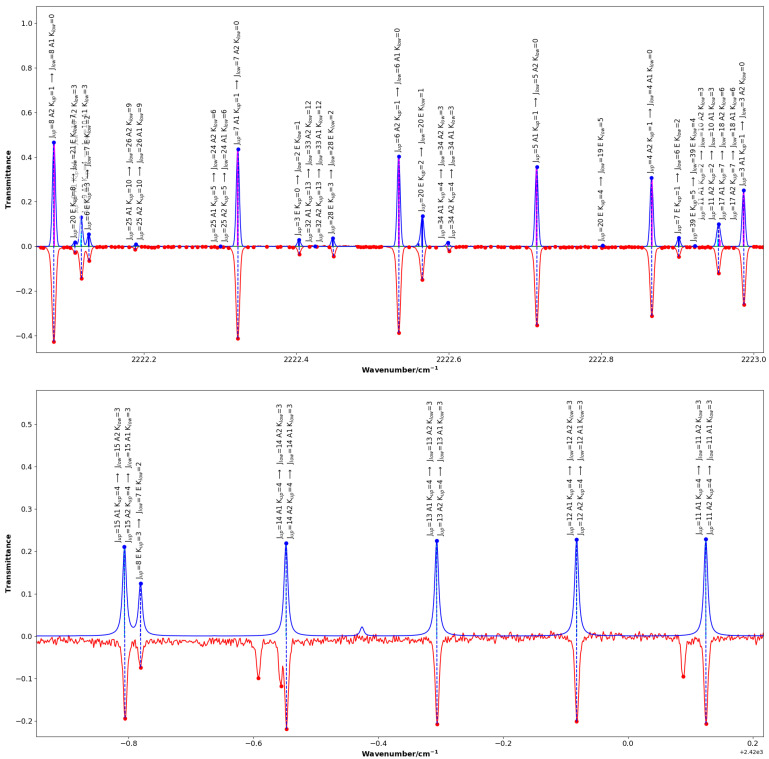
Same as Figure 2, for the $\nu_{3}+\nu_{6}$ band (**upper panel**), and the $2\nu_{6}$ band (**lower panel**) where several lines belonging to the $\nu_{6}+\nu_{2}$ band are also visible in the experimental spectrum (in red) which are not reproduced in the synthetic spectrum (in blue). Calculated line intensities are displayed as black sticks within the profiles, with K=0 transitions in magenta and K=3p in blue.

**Figure 4 molecules-30-04389-f004:**
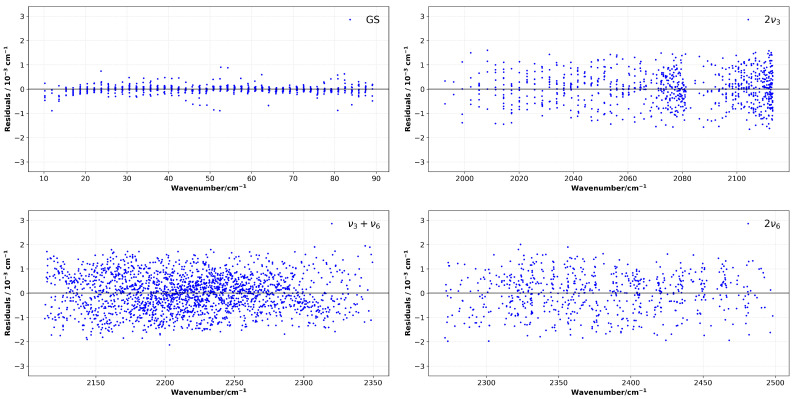
Residuals between observed and calculated line positions for the ground state and the triad. The vertical scale is fixed to ±4 × 10^−3^ cm^−1^ for all panels to allow direct visual comparison.

**Figure 5 molecules-30-04389-f005:**
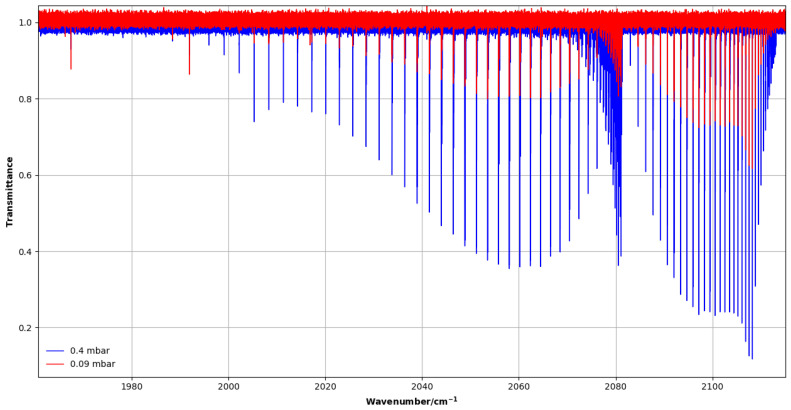
High-resolution transmittance spectra of the 2$\nu_{3}$ band of CH_3_F recorded with 0.09 and 0.4 mbar pressures in the 1950–2120 cm^−1^ region.

**Figure 6 molecules-30-04389-f006:**
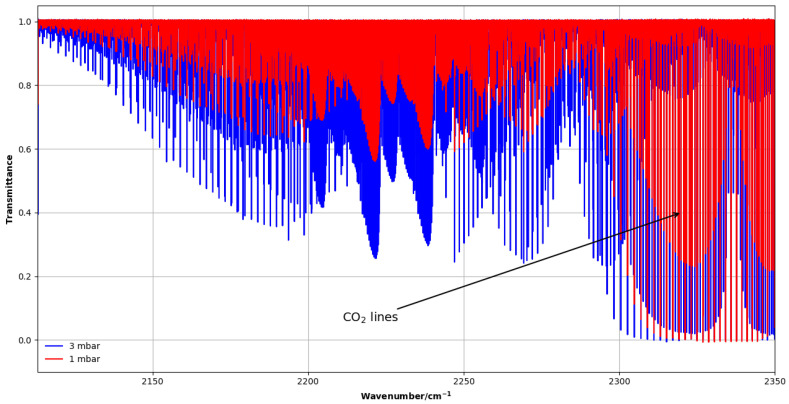
Overview of CH_3_F experimental spectra of the $\nu_{3}+\nu_{6}$ band of CH_3_F in the 2100–2350 cm^−1^ region at 1 and 3 mbar, showing strong CO_2_ absorption lines.

**Figure 7 molecules-30-04389-f007:**
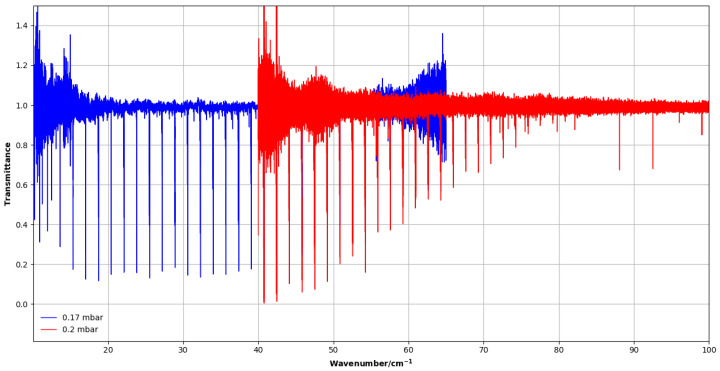
Far-infrared spectra of CH_3_F recorded at SOLEIL. The blue spectrum was taken at 0.17 mbar and was measured with the 1.6 K bolometer. The red spectrum was recorded at 0.2 mbar with the 4 K bolometer. Overlap of spectra between two measurements provides a consistent scale of intensity and full coverage of the 10–100 cm^−1^ range.

**Table 1 molecules-30-04389-t001:** Experimental conditions for CH_3_F spectra recorded at GSMA, Reims.

Band	Range (${\mathrm{cm}}^{-1}$)	Path Length (m)	Pressures (mbar)	Resolution (${\mathrm{cm}}^{-1}$)
2$\nu_{3}$	1950–2100	3.2	0.09, 0.40	0.003
$\nu_{3}+\nu_{6}$	2100–2350	8.2	1.00, 3.00	0.003
2$\nu_{6}$	2300–2450	38.6	1.00, 2.56	0.003

**Table 2 molecules-30-04389-t002:** Experimental conditions for CH_3_F spectra recorded at SOLEIL synchrotron.

Region	Range (${\mathrm{cm}}^{-1}$)	Detector	Pressures (mbar)	Resolution (${\mathrm{cm}}^{-1}$)
Rotational FIR	10–40	Bolometer (1.6 K)	0.07–0.17	0.001
Rotational FIR	40–100	Bolometer (4 K)	0.21–0.59	0.001

**Table 3 molecules-30-04389-t003:** Sample of Electronic Appendix A. Observed methyl fluoride transitions with assignments in the ground state and the $\nu_{3}+\nu_{6}$ regions. $\nu_{0}$ corresponds to the measured line positions (cm^−1^). The rotational assignment is given in terms of the vibrational polyad number *P*, the rotational quantum number *J*, the rovibrational symmetry type Γ, the rovibrational ranking index α, and the rotational quantum number *K* for the lower and upper states. The vibrational assignment contains the principal vibrational quanta and the vibrational symmetry type of both states. $E_{\mathrm{low}}$ is the lower state energy (cm^−1^). Obs-Calc corresponds to the difference between the observed and calculated line positions (in 10^−3^ cm^−1^). The last column indicates the type of assignment (automatic or manual).

$\nu_{0}$ (cm^−1^)	Rotational Assignment										Vibrational Assignment				$E_{\mathrm{low}}$ (cm^−1^)	Obs–Calc (10^−3^ cm^−1^)	Type
	Lower State					Upper State					Lower		Upper				
79.0393	0	46	$A_{1}$	5	12	0	47	$A_{2}$	5	12	000000	$A_{1}$	000000	$A_{1}$	2449.780201	−0.05	Auto
79.0393	0	46	$A_{2}$	4	12	0	47	$A_{1}$	4	12	000000	$A_{1}$	000000	$A_{1}$	2449.780201	−0.05	Auto
79.0705	0	46	E	8	11	0	47	E	8	11	000000	$A_{1}$	000000	$A_{1}$	2351.326959	0.42	Auto
79.0981	0	46	E	7	10	0	47	E	7	10	000000	$A_{1}$	000000	$A_{1}$	2261.371357	−0.09	Auto
79.1237	0	46	$A_{1}$	4	9	0	47	$A_{2}$	4	9	000000	$A_{1}$	000000	$A_{1}$	2179.930717	0.01	Auto
79.1237	0	46	$A_{2}$	3	9	0	47	$A_{1}$	3	9	000000	$A_{1}$	000000	$A_{1}$	2179.930717	0.01	Auto
79.1464	0	46	E	6	8	0	47	E	6	8	000000	$A_{1}$	000000	$A_{1}$	2107.020710	−0.13	Auto
79.1666	0	46	E	5	7	0	47	E	5	7	000000	$A_{1}$	000000	$A_{1}$	2042.655358	−0.12	Auto
79.1842	0	46	$A_{1}$	3	6	0	47	$A_{2}$	3	6	000000	$A_{1}$	000000	$A_{1}$	1986.847033	−0.05	Auto
79.1842	0	46	$A_{2}$	2	6	0	47	$A_{1}$	2	6	000000	$A_{1}$	000000	$A_{1}$	1986.847033	−0.05	Auto
79.1991	0	46	E	4	5	0	47	E	4	5	000000	$A_{1}$	000000	$A_{1}$	1939.606458	0.00	Auto
79.2112	0	46	E	3	4	0	47	E	3	4	000000	$A_{1}$	000000	$A_{1}$	1900.942707	−0.05	Auto
79.2206	0	46	$A_{1}$	2	3	0	47	$A_{2}$	2	3	000000	$A_{1}$	000000	$A_{1}$	1870.863202	−0.12	Auto
79.2206	0	46	$A_{2}$	1	3	0	47	$A_{1}$	1	3	000000	$A_{1}$	000000	$A_{1}$	1870.863202	−0.12	Auto
79.2274	0	46	E	2	2	0	47	E	2	2	000000	$A_{1}$	000000	$A_{1}$	1849.373717	−0.08	Auto
79.2315	0	46	E	1	1	0	47	E	1	1	000000	$A_{1}$	000000	$A_{1}$	1836.478376	−0.04	Auto
79.2329	0	46	$A_{1}$	1	0	0	47	$A_{2}$	1	0	000000	$A_{1}$	000000	$A_{1}$	1832.179655	0.00	Auto
2222.0819	0	8	$A_{1}$	1	0	3	8	$A_{2}$	2	1	000000	$A_{1}$	001001	E	61.318769	−0.24	Auto
2222.1095	0	21	E	5	7	3	20	E	27	8	000000	$A_{1}$	001001	E	604.781092	0.34	Auto
2222.1182	0	10	$A_{1}$	2	3	3	9	$A_{2}$	6	4	000000	$A_{1}$	001001	E	132.624744	0.32	Auto
2222.1182	0	10	$A_{2}$	1	3	3	9	$A_{1}$	5	4	000000	$A_{1}$	001001	E	132.624744	0.32	Auto
2222.1281	0	7	E	2	2	3	6	E	8	3	000000	$A_{1}$	001001	E	65.010605	0.20	Auto
2222.1883	0	26	$A_{1}$	4	9	3	25	$A_{2}$	19	10	000000	$A_{1}$	001001	E	946.427107	−1.02	Auto
2222.1883	0	26	$A_{2}$	3	9	3	25	$A_{1}$	17	10	000000	$A_{1}$	001001	E	946.427107	−1.02	Auto
2222.3232	0	7	$A_{2}$	1	0	3	7	$A_{1}$	2	1	000000	$A_{1}$	001001	E	47.694176	−0.15	Auto
2222.4037	0	2	E	1	1	3	3	E	3	0	000000	$A_{1}$	001001	E	9.440744	−0.04	Auto
2222.4489	0	28	E	2	2	3	28	E	8	3	000000	$A_{1}$	001001	E	707.604202	1.04	Auto
2222.5344	0	6	$A_{1}$	1	0	3	6	$A_{2}$	2	1	000000	$A_{1}$	001001	E	35.771813	−0.04	Auto
2222.5600	0	14	E	3	4	3	15	E	9	3	000000	$A_{1}$	001001	E	248.004363	−0.22	Manu
2222.7154	0	5	$A_{2}$	1	0	3	5	$A_{1}$	2	1	000000	$A_{1}$	001001	E	25.552018	−0.01	Auto
2222.8004	0	19	E	4	5	3	20	E	11	4	000000	$A_{1}$	001001	E	431.464091	−0.82	Manu
2222.8663	0	4	$A_{1}$	1	0	3	4	$A_{2}$	2	1	000000	$A_{1}$	001001	E	17.035080	0.05	Auto
2222.9014	0	6	E	2	2	3	7	E	6	1	000000	$A_{1}$	001001	E	53.089063	−0.25	Auto
2222.9224	0	39	E	3	4	3	39	E	12	5	000000	$A_{1}$	001001	E	1392.808779	−0.48	Auto
2222.9537	0	18	$A_{1}$	3	6	3	17	$A_{2}$	12	7	000000	$A_{1}$	001001	E	446.695134	−1.56	Auto
2222.9537	0	18	$A_{2}$	2	6	3	17	$A_{1}$	10	7	000000	$A_{1}$	001001	E	446.695134	−1.23	Auto
2222.9537	0	10	$A_{1}$	2	3	3	11	$A_{2}$	5	2	000000	$A_{1}$	001001	E	132.624744	0.93	Auto
2222.9537	0	10	$A_{2}$	1	3	3	11	$A_{1}$	4	2	000000	$A_{1}$	001001	E	132.624744	0.93	Auto
2222.9871	0	3	$A_{2}$	1	0	3	3	$A_{1}$	2	1	000000	$A_{1}$	001001	E	10.221241	0.18	Auto

**Table 4 molecules-30-04389-t004:** Results of the effective Hamiltonian fits obtained in this work (TW) and by Khan et al. [18]. $J_{\max}$ and $K_{\max}$ correspond to the maximum rotational quantum numbers used in the fit of the Hamiltonian parameters. *N* denotes the number of assigned transitions included in the fit. RMS represents the standard deviation of the residuals between the observed and calculated line positions.

Polyad	State	Vib. Energy (cm^−1^)	$J_{\max}/K_{\max}$		*N*		RMS (10^−4^ cm^−1^)	
			Khan	TW	Khan	TW	Khan	TW
Ground state	GS	0	–/–	70/16	–	849	–	3.55
Dyad 1	$\nu_{3}=1;\;A_{1}$	1048.6	44/18	–/–	564	–	6.75	–
	$\nu_{6}=1;\;E$	1182.6	28/12	–/–	784	–	5.06	–
Dyad 2	$\nu_{2}=1;\;A_{1}$	1459.3	34/6	–/–	231	–	8.34	–
	$\nu_{5}=1;\;E$	1467.8	37/13	–/–	1385	–	28.1	–
Triad	$\nu_{3}=2;\;A_{1}$	2081.3	27/11	45/19	119	1092	9.11	6.91
	$\nu_{3}=1,\;\nu_{6}=1;\;E$	2221.8	24/23	45/17	105	2532	9.42	6.85
	$\nu_{6}=2;\;E$	2365.9	–/–	39/9	–	696	–	7.86
	$\nu_{6}=2;\;A_{1}$	2359.4	–/–	37/13	–	135	–	8.99

## Data Availability

The original contributions presented in this study are included in the article and its Appendix A. Further inquiries can be directed to the corresponding author.

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
