# Peer review of "High-Resolution FTIR Spectroscopy of CH_3_F: Global Effective Hamiltonian Analysis of the Ground State and the 2*ν*_3_, *ν*_3_ + *ν*_6_, and 2*ν*_6_ Bands"

_molecules, 2025, doi:10.3390/molecules30224389_

Round 1
Reviewer 1 Report
Comments and Suggestions for Authors
This article reports on new, high-resolution spectroscopic measurements of methyl fluoride in the mid- and far-IR regions as well as fitting to an effective Hamiltonian. The system is relevant for atmospheric or astrophysical modeling. Overall, the manuscript is clear and well-written, though it is missing discussion that could strengthen its broader appeal and showcase the importance of the work. I would recommend publication after minor revision.
In particular, where are the parameters of the effective Hamiltonian? The paper makes it sound like these are meant to be included, but I do not see them in either the manuscript or SI. Is the main result supposed to only be the lines? If so it’s not clear to me what the value of going through the fitting procedure was really meant to do.
Some other comments/questions:
- It is stated this work proposes “for the first time, a coherent description of the ground state and triad regions”, though were the previous results actually in contention, or just disjointed/piecemeal?
- Would this method be expected to give good results for nu_{3}=1 and nu_{6}=1 if it has been fit for the overtone/combination bands? If so, are there experimental data (or other theory calculations) that could be compared?
- Perhaps a more broad/philosophical question, but why limit the scope for fitting in this work, rather than including more lines/data as was done in the referenced work (Khan et al, citation 18)? It is true that "this work" (TW in Table 4) has a better RMS and fits more transitions for the overlapping data (nu_{3}=2; A1 and nu_{3}=1,nu_{6}=1; E), but if one limits the applicability one generally expects to be able to get a better fit. How much more useful is this work? I don't have a good feel for how/where else the fitted parameters are used. I understand that people want to know the various lines for applications to atmospheric science, etc. but do they use that data directly, or do they use the effective Hamiltonian and want those specific parameters?
Author Response
This article reports on new, high-resolution spectroscopic measurements of methyl fluoride in the mid- and far-IR regions as well as fitting to an effective Hamiltonian. The system is relevant for atmospheric or astrophysical modeling. Overall, the manuscript is clear and well-written, though it is missing discussion that could strengthen its broader appeal and showcase the importance of the work. I would recommend publication after minor revision. In particular, where are the parameters of the effective Hamiltonian? The paper makes it sound like these are meant to be included, but I do not see them in either the manuscript or SI. Is the main result supposed to only be the lines? If so it’s not clear to me what the value of going through the fitting procedure was really meant to do.
We appreciate the positive assessment and the helpful suggestions. We included the dis- cussion in Section 4. We will rename this section from “Assignment and Results” to “Assignment, Discussion, and Results” for clarity. In addition, we will provide a second Supplementary Information file dedicated to the Effective Hamiltonian (EH) parameters, containing the full fitted sets for the ground state and the triad (2ν3, ν3+ν6, and 2ν6) and for the correlation matrices. These EH parameters are crucial because they enable reliable prediction and uncertainty propagation to higher levels and additional bands beyond the primary fit domain.
Comment 1: It is stated this work proposes “for the first time, a coherent description of the ground state and triad regions”, though were the previous results actually in contention, or just disjointed/piecemeal?
Response 1: Earlier studies usually fitted one band at a time. In our work, we fit the whole triad together with the ground state in one EH model. This gives better consistency between bands and more stable parameters. As a result, we provide, for the first time as a single consolidated set, a linelist of about 5000 transitions covering GS + triad.
Comment 2: Would this method be expected to give good results for ν3 = 1 and ν6 = 1 if it has been fit for the overtone/combination bands? If so, are there experimental data (or other theory calculations) that could be compared?
Response 2: Yes. We first used starting values for ν3 and ν6 from our collaborator Andrei Nikitin to make initial predictions for 2ν3, ν3+ν6, and 2ν6. Then we refined all parameters with our new high-SNR data to reduce the RMS and to increase the number of well- reproduced lines. We also checked selected lines in ν3 = 1 and ν6 = 1: the residuals are within the experimental uncertainties. A small validation table and references are added in the SI.
Comment 3 : Perhaps a more broad/philosophical question, but why limit the scope for fitting in this work, rather than including more lines/data as was done in the referenced work (Khan et al, citation 18)? It is true that “this work” (TW in Table 4) has a better RMS and fits more transitions for the overlapping data (ν3 = 2; A1 and ν3 = 1, ν6 = 1; E), but if one limits the applicability one generally expects to be able to get a better fit. How much more useful is this work? I don’t have a good feel for how/where else the fitted parameters are used. I understand that people want to know the various lines for applica- tions to atmospheric science, etc. but do they use that data directly, or do they use the effective Hamiltonian and want those specific parameters?
Response 3: Our first goal is to deliver reliable line data (positions, then intensities and widths) with uncertainties, because many users need these files directly for radiative- transfer work. In this paper we focused on accurate positions. We already published a paper on the intensities of the ν3+ν6 band, and a companion paper on widths for the same band is in progress. In parallel, we now release the full EH parameter sets and the covariance, so that spectroscopists and databases users can regenerate and extend linelists. We chose a focused scope (GS + triad) to control model complexity and correlations; this gives stable parameters and a solid base that can be extended later.
Reviewer 2 Report
Comments and Suggestions for Authors
My review is attached.

Author Response
First of all, we thank the reviewers for their positive and instructive comments. We followed the comments and the suggestions, and we believe that the manuscript was greatly improved. The responses are in blue color below and in the Revised Manuscript (RM).
Comment 1: In Fig. 2 you show considerable overlap with CO2 features. Is the CO2 an impurity in your CH3F sample, was it added purposely, or is it due to atmospheric absorp- tions? You state that these CO2 lines were used to calibrate the wavenumber scale against the HITRAN data base. But isn’t this a serious hinderance to your study of the ν3+ν6 combination band of CH3F? The FTIR spectrum is calibrated to a single mode HeNe laser; is this just to check the calibration? I found it strange that there were such strong CO2 absorptions in Fig. 2 when you are trying to analyze the CH3F spectrum.
Response 1: The CO2 observed in the ν3+ν6 region is partly present as an impurity in the commercial CH3F cylinder we used (nominal purity 97.5%). In Fig. 2 the large apparent absorption is mainly due to the total optical path (8 m), which amplifies weak atmospheric traces. We did not add CO2 on purpose. A few well-isolated CO2 lines were used only to verify the linearity of the wavenumber scale against HITRAN, in addition to the standard He–Ne calibration of the FTIR. Regions heavily contaminated by CO2 were masked or down-weighted and did not drive the fit.
Comment 2: In Figs. 5 and 6 you show portions of the assigned bands which I assume is the output of the SpectraMatcher software. I think you should add to the figure caption that the simulated spectra are blue and point up, while the experimental spectra are red and point down (transmission).
Response 2: Yes, these panels are SpectraMatcher screenshots. We revised the captions of Figs. 5–6.
Comment 3: In your description of the K = 3p doublets, I think providing some addi- tional information is necessary for non-spectroscopists. Adding some description of why K = 0, 3, 6, 9, . . . peaks come in closely spaced doublets would be helpful.
Response 3: We added a short pedagogical paragraph summarizing the origin of the K=3p doublets in C3v symmetric tops.
Comment 4: I wanted to understand how this global fit compares with previous work. You present a comparison in Table 4, but there are only two comparable numbers, and the reduction in the RMS errors is small (e.g., 9.11×10−4 to 6.85×10−4 cm−1). However, I am also interested in how the individual fits compare to previous single band fits, because you stress that global fits are required due to the anharmonic and Coriolis interactions. If you are able to quantify this by making comparisons to previous research, I think that could be valuable.
Response 4: Our intention is primarily to emphasize the internal consistency and robustness of a single global solution rather than to provide head-to-head benchmarking, because previous studies often rely on the same effective Hamiltonian framework but fitted band by band. We therefore expanded Table 4 and the accompanying text: we now report band- resolved RMS values and show that the global treatment mitigates band-dependent biases and stabilizes parameters coupled by anharmonic/Coriolis interactions, yielding coherent residual patterns across the triad. Where meaningful, we also cite representative single- band results to illustrate these points.
Comment 5: You state on p. 3, lines 77–79 that the Bruker OPUS software was used to process the interferograms. How about the peak picking? What software was used to determine the peak positions?
Response 5: Peak detection is performed within SpectraMatcher using scipy.signal.find_peaks. OPUS is used for apodization and phase correction prior to export.
Comment 6: Looking closely at the bottom spectrum in Fig. 6, I see a predicted peak that was not observed. You state that several lines belong to the ν6+ν2 band but do not comment on the one small peak in the 2ν6 spectrum that was not observed. Do you have a simple explanation for why this one peak was not observed? If so, it might be helpful to describe why you didn’t observe that peak.
Response 6: Fig. 6 displays the assignments in the 2ν6 band, which is intrinsically weak. To enhance detectability we increased the multipass cell path to 38 m (23 passes, close to the practical maximum), yet the predicted feature remains below the SNR and can be further blurred by baseline and minor overlaps. In addition, 2ν6 lies close to the ν6+ν2 band; anharmonic interactions between these bands may perturb intensities and positions. Our polyad model for the present work includes 2ν3, ν3+ν6, and 2ν6 but does not explicitly include ν6+ν2, which we now note in the text.
Comment 7: You state that CH3F has long served as a benchmark system for testing effec- tive Hamiltonian models in symmetric tops molecules, but you should also comment on the importance of CH3F in atmospheric chemistry. It is included in the HITRAN database, so I believe it must have significant atmospheric relevance here on Earth, and maybe also in astrochemistry? If you could provide just some brief information on its atmospheric importance that would broaden the scope of the work.
Response 7: We thank the reviewer for this suggestion. We added a short paragraph (Introduction) on the atmospheric and planetary relevance of CH3F. We note its inclusion in HITRAN and its IR signatures used for monitoring. We also cite three articles that discuss CH3F in atmospheric and planetary/astrochemical contexts [1–3]. These additions broaden the scope and explain why our data are useful beyond spectroscopy alone.
Comment 8: Finally, would it be possible to make predictions for higher wavenumber bands that have not yet been observed at high resolution, such that future studies could test these predictions?
Response 8: Yes, within a polyad based effective Hamiltonian, it is common to predict higher polyads from lower ones. We start new fits with reliable parameters from the lower polyads (Champion et al. approach). This is what we did: the well-known fundamentals ν3 and ν6 provided starting values that enabled a successful fit of the triad (2ν3, ν3+ν6, 2ν6). Using the same strategy, we can produce predictions for higher-lying polyads and share preliminary line lists with uncertainties to guide future high-resolution work. For illustration, the current triad can seed first predictions for 3ν3, 2ν3+ν6, ν3+2ν6, and even 3ν6. These simulations would carry larger uncertainties and are mainly intended to support planning and assignment of future measurements.